# Clinically applicable histopathological diagnosis system for gastric cancer detection using deep learning

Zhigang Song[1,9], Shuangmei Zou[2,9], Weixun Zhou[3,9], Yong Huang[1], Liwei Shao[1], Jing Yuan[1], Xiangnan Gou[1], Wei Jin[1], Zhanbo Wang[1], Xin Chen[1], Xiaohui Ding[1], Jinhong Liu[1], Chunkai Yu[4], Calvin Ku[5], Cancheng Liu[5], Zhuo Sun[5], Gang Xu[5], Yuefeng Wang[5], Xiaoqing Zhang[5], Dandan Wang[6], Shuhao Wang [5,7✉], Wei Xu[7], Richard C. Davis[8] & Huaiyin Shi [1✉]

The early detection and accurate histopathological diagnosis of gastric cancer increase the chances of successful treatment. The worldwide shortage of pathologists offers a unique opportunity for the use of artificial intelligence assistance systems to alleviate the workload and increase diagnostic accuracy. Here, we report a clinically applicable system developed at the Chinese PLA General Hospital, China, using a deep convolutional neural network trained with 2,123 pixel-level annotated H&E-stained whole slide images. The model achieves a sensitivity near 100% and an average specificity of 80.6% on a real-world test dataset with 3,212 whole slide images digitalized by three scanners. We show that the system could aid pathologists in improving diagnostic accuracy and preventing misdiagnoses. Moreover, we demonstrate that our system performs robustly with 1,582 whole slide images from two other medical centres. Our study suggests the feasibility and benefits of using histopathological artificial intelligence assistance systems in routine practice scenarios.

[1] Department of Pathology, The Chinese PLA General Hospital, 100853 Beijing, China. [2] Department of Pathology, National Cancer Center/National Clinical Research Center for Cancer/Cancer Hospital, Chinese Academy of Medical Sciences and Peking Union Medical College, 100021 Beijing, China. [3] Department of Pathology, Peking Union Medical College Hospital, 100005 Beijing, China. [4] Department of Pathology, Beijing Shijitan Hospital, Capital Medical University, 100038 Beijing, China. [5] Thorough Images, 100102 Beijing, China. [6] Department of Pathology, Third Hospital, School of Basic Medical Sciences, Peking University Health Science Center, 100083 Beijing, China. [7] Institute for Interdisciplinary Information Sciences, Tsinghua University, 100084 Beijing, China. [8] Department of Pathology, Duke University Medical Center, Durham, NC 27710-1000, USA. [9] These authors contributed equally: Zhigang Song, Shuangmei Zou, Weixun Zhou. ✉email: ericwang@tsinghua.edu.cn; shihuaiyin@sina.com

Gastric cancer is the fifth most common cancer worldwide and the third leading cause of cancer death[1]. There is a wide geographic difference in its incidence, with the highest incidence rate in East Asian populations[2]. In China, ~498,000 new cases of gastric cancer were diagnosed in 2015, and it was the second leading cause of cancer-associated mortality[3]. As early detection, accurate diagnosis, and surgical intervention are crucial factors for reducing gastric cancer mortality, robust and consistently effective pathology services are indispensable. However, there is a critical shortage of anatomical pathologists both nationally and globally, which has created overloaded workforces, thus effecting diagnostic accuracy[4]. An increasing number of pathology laboratories have adopted digital slides in standard practice in the form of whole slide images (WSIs) in daily routine diagnostics[5–7]. The transformation of practice from microscope to WSI has paved the way for using artificial intelligence (AI) assistance systems in pathology to overcome human limitations and reduce diagnostic errors. This has allowed the development of innovative approaches, such as AI via deep learning[8–18]. Studies have focused on developing algorithms that can flag the suspicious areas, prompting pathologists to examine the tissue thoroughly under high magnification or employ immunohistochemical (IHC) studies when necessary and make an accurate diagnosis[19].

While recent studies have validated the effectiveness of pathology AI for tumor detection in various organ systems, such as lung[20], stomach[21], lymph node metastases in breast cancer[22–24], prostate core needle biopsies[24–26], and mesothelioma[27], we identify many nontrivial challenges that should be addressed before considering application in the clinical setting. First, a deep learning model should be able to sustain a thorough test with a substantial number (i.e., thousands) of slides over a continuous time period and with WSIs procured by various brands of digital scanners. The sensitivity should be near 100% without compromising specificity too heavily. Second, with the assistance of the AI system, pathologists should be able to improve their diagnostic accuracy while not drawing out the routine reporting process. To further boost the trust of pathologists in AI assistance systems, the model predictions should be investigated to determine their strengths and weaknesses. Finally, it is necessary to conduct a multicentre test before system deployment to guarantee the stability of the model performance across different hospitals. Previous studies have addressed some of these challenges, but none have met all these criteria.

Here, we report the latest operation of the AI assistance system at the Chinese PLA General Hospital (PLAGH), China, with careful consideration of the solutions to the challenges that we discussed above. The deep learning model is trained with 2123 pixel-level annotated haematoxylin and eosin (H&E)-stained digital slides from 1500 patients, which include 958 surgical specimens (908 malignancies) and 542 biopsies (102 malignancies) with diverse tumor subtypes; details are illustrated in Fig. 1a (abbreviations are given in Supplementary Table 1). The training slides are produced at ×40 magnification (0.238 μm/pixel) by the National Medical Products Administration-cleared KFBio KF-PRO-005 digital scanner. We develop an iPad-based annotation system and provide a standard operating procedure (SOP) for data collection and annotation to 12 senior pathologists (see Supplementary Table 2). We adopt the 4th edition of the WHO Classification of Tumors of the Digestive System as the reference standard[28]. The pathologists circle the precise areas using the Apple Pencil with preset labels including malignant, benign, poor quality, and ignore (see Supplementary Fig. 1 for several labeled samples). We assign the malignant label to both high-grade intraepithelial neoplasia and carcinoma because both lesions require surgical intervention. Labels of poor quality and ignore

are assigned to areas with low preparation or scanning quality and slides difficult to diagnose, respectively.

We utilize a convolutional neural network (CNN) of DeepLab v3 architecture for our binary image segmentation approach, which enables pixel-level cancer detection. The WSIs and their corresponding annotations are split into 320 × 320-pixel patches at ×20 magnification (0.476 μm/pixel) and then feed into the network for training. We perform carefully designed data augmentation during training. Since histopathological slides have no specific orientation, we apply random rotations by 90, 180, and 270 degrees and random flips (horizontal and vertical) to the training patches. To boost the model stability for WSIs collected from different hospitals and digitalized from various scanners, we also apply Gaussian and motion blurs and color jittering in brightness (0.0–0.2), saturation (0.0–0.25), contrast (0.0–0.2), and hue (0.0–0.04). During training, we consider 'poor quality' as 'ignore,' and neglect losses coming from the 'ignore' class. In the inference phase, each pixel is assigned a probability of being malignant by the trained model. Slide-level prediction is obtained by sorting the probabilities of all pixel-level predictions. We adopt the top 1000 probabilities and use the mean to represent the slide-level prediction (a detailed comparison of slide-level predictors is provided in Supplementary Table 3). Compared with the commonly adopted approaches that utilize patch classification and sliding windows[29,30], the semantic segmentation approach[31–33] gives a more detail-rich prediction at the pixel level (see Supplementary Table 4 and Supplementary Fig. 2).

The AI assistance system achieves a sensitivity of 0.996 and an average specificity of 0.806 on the daily gastric dataset from PLAGH with 3212 WSIs digitalized with 3 scanner models. We show that with the assistance of the system, pathologists improve diagnostic accuracy, and reduce misdiagnoses. Furthermore, the multicentre test with 1582 WSIs from 2 other medical centers confirms the robustness of the system.

## Results

**Trial run.** The AI assistance system was deployed in PLAGH and underwent a 3-month (June 2017 to August 2017) trial run with the daily gastric dataset. Overall, 3212 daily gastric slides from 1814 patients (1101/713 males/females with average ages of 54.12/54.66 years, see Supplementary Fig. 3 for detailed distribution) included 154 surgical specimens (118 malignancies) and 1660 biopsies (61 malignancies). The slides were grouped biweekly and divided into six consecutive time periods. To test the model performance on data produced by different scanners (see Supplementary Fig. 4), the slides were digitalized by three scanner models, including KFBio KF-PRO-005 (403 WSIs, ×40, 0.238 μm/pixel), Ventana DP200 (977 WSIs, ×40, 0.233 μm/pixel), and Hamamatsu NanoZoomer S360 (1832 WSIs, ×40, 0.220 μm/pixel). With this dataset, the model revealed a stable performance with an average area under the curve (AUC) of 0.986 (accuracy: 0.873, sensitivity: 0.996, specificity: 0.843) and a standard deviation of 0.018 (0.099, 0.011, 0.109) across the timeline, as shown in Fig. 1b. The detailed receiver operating characteristic (ROC) curves are provided in Supplementary Fig. 5, see Supplementary Fig. 6 for four examples of predicted heatmaps. The sensitivities of tubular adenocarcinoma and poorly cohesive carcinoma were 0.998 and 1.0, respectively, excluding mixed adenocarcinoma. We compared how the model performed on the WSIs produced by the three scanners, as shown in Fig. 1c. Compared with KFBio KF-PRO-005, we observed slight model performance drops, with AUC (accuracy, sensitivity, specificity) of 0.004 (0.032, 0.005, 0.040) and 0.013 (0.170, 0.0, 0.210) on Ventana DP200 and Hamamatsu NanoZoomer S360, respectively (detailed results listed in Supplementary Table 5).

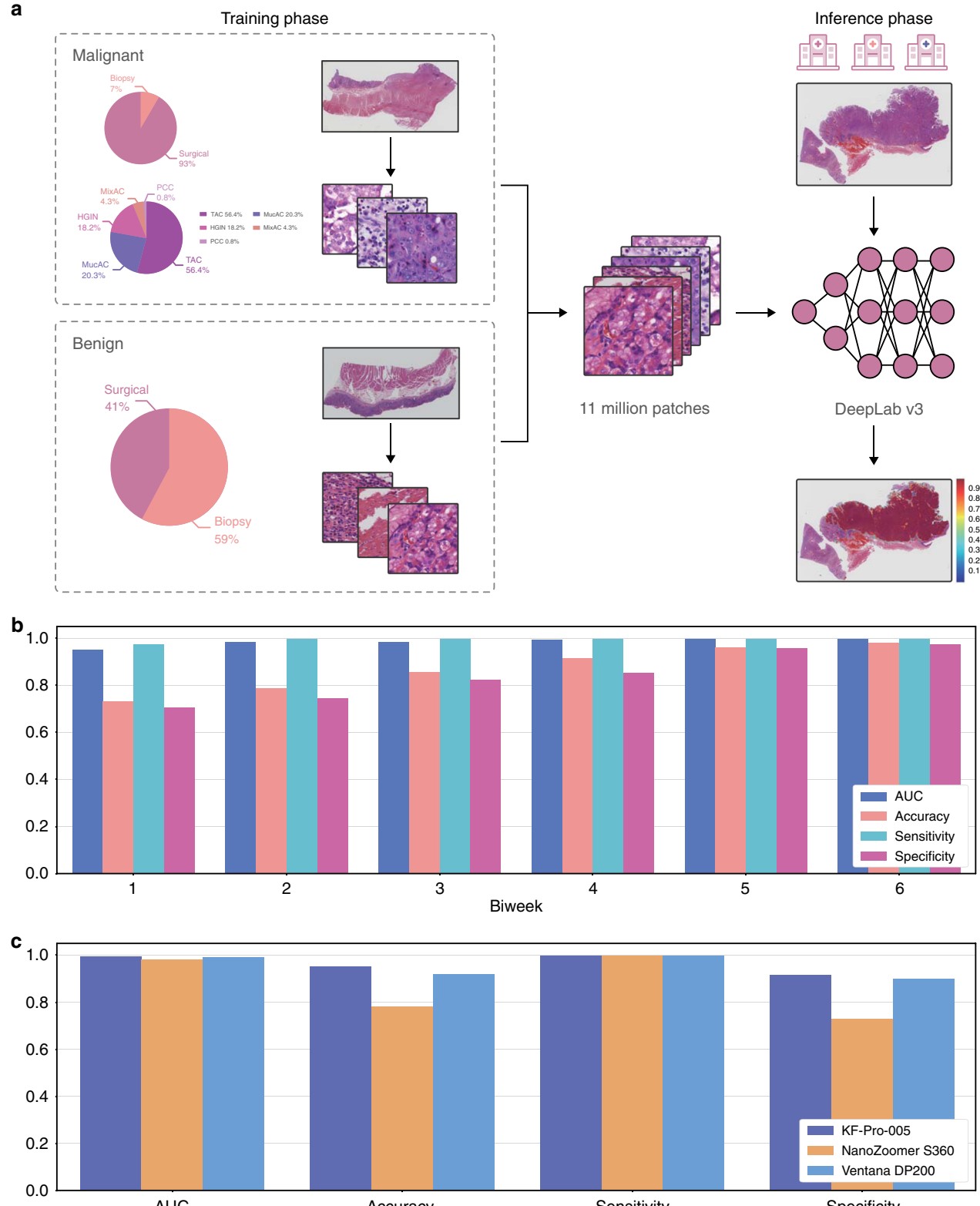

**Fig. 1 The framework of our research and model performance on the daily gastric dataset. a** Deep learning model training and inference. We trained the model using WSIs digitalized and annotated at PLAGH. We illustrated the training data distribution at the slide level. The abbreviations are detailed in Supplementary Table 1. The trained model was tested by slides collected from PLAGH and two other hospitals. **b** The plot of the model performance histogram of the slides from the daily gastric dataset. **c** Model performance histogram of the daily gastric slides digitalized by three different scanners.

**Assistance for pathologists**. To prevent overlooking malignancy, the AI assistance system should be able to highlight abnormal areas, prompting pathologists to perform a scrutinized reassessment. During the daily gastric slide examination, we found two missed cases that were overlooked in the initial reports and caught by the AI assistance system. The first case had received disparate diagnoses from the biopsy report and the surgical report, respectively. Cancer was found in the surgical specimen and reported in the surgical report, but because the cancer cells were limited in number, they were missed in the initial biopsy report. In the slide, the tumor cells were scattered under the normal foveolar epithelium and only better visible under high magnification, as shown in Fig. 2a(i). The other case shown in Fig. 2a(ii) contained deceptively bland-looking cancer cells, reflecting another example that can easily be missed. Nevertheless, in the AI-predicted heatmap, several red dots clearly marked the positions of the malignant tumor cells. These kinds of misdiagnoses are uncommon but possible, especially when a case is read in haste, such as the last case of the day or slides read while multitasking. The AI assistance system successfully flagged these subtle regions, which indicated that it may alert pathologists to re-examine the slides and/or perform ancillary tests in a real-world scenario.

The current AI assistance system could function not only as a preanalytic tool to prioritize early attention to suspicious cases for review but also as an analog to a second opinion from fellow pathologists. For difficult cases, especially for slides advised to have additional IHC stains, the model prediction had a noticeable influence on the final diagnosis. We created an IHC dataset with all the H&E-stained slides in the daily gastric dataset that were subjected to IHC examination. The IHC dataset contained 27 surgical specimens (20 malignancy) and 72 biopsies (22 malignancies). Our model achieved an AUC of 0.923 (accuracy: 0.808, sensitivity: 0.976, specificity: 0.684). In Fig. 2b, we observed a clear segregation of confidence in the model between malignant and benign cases. The model performance was reasonably accurate with the malignant cases, while it showed less confidence with the benign cases as the distribution spread out with significantly larger variance. While this model made predictions only based on H&E-stained WSIs, we demonstrated that our model could provide a useful visual cue using a heatmap along with providing a cancer risk probability. On the left side of Fig. 2b, we also showed benign cases sorted by probability for comparison with malignant cases. We observed that the benign cases given low cancer probabilities by the model were those with clearer visual cues and could be diagnosed without resorting to IHC, whereas those with higher cancer probabilities were the more challenging cases, which required scrutinized examination under low and high magnifications and sometimes ancillary tests.

**Internal examination**. To test whether our AI assistance system was able to make an accurate diagnosis in real-world scenarios, we conducted an examination using 100 slides to assess the performance of 12 junior pathologists who were under training. As shown in Fig. 3a, 100 slides were categorized into four groups depending on the degree of diagnostic difficulty: (I) easy to diagnose under low magnification (34 WSIs); (II) easy to diagnose but needed examination under high magnification (39 WSIs); (III) difficult to diagnose, ancillary IHC not required (23 WSIs); and (IV) challenging to diagnose, required ancillary IHC (4 WSIs). We randomly divided the pathologists into three groups: a microscopy group, a WSI group, and an AI-assisted group. As the names suggested, the microscopy group worked with microscopes, the WSI group with WSIs, and the AI-assisted group with digital slides plus the AI assistance system. The examination was carried out in duplicate

with a 1-h time constraint and without time constraints. In Fig. 3b, we compared the performance of the pathologists with the model prediction performance using the ROC curve. We observed that the model performance was on par with the performance of the human pathologists, even exceeding the average performance of the 12 pathologists. We discovered that the AI assistance system helped the pathologists achieve better accuracy, as shown in Fig. 3c. With the help of the system, the average accuracy increased by 0.008/0.060 and 0.013/0.018 compared with the microscopy and WSI groups with/without time constraints, respectively. In addition to the improvement in diagnostic accuracy, the AI assistance system was able to assist the pathologists in performing more consistently, even under a time constraint. When comparing the diagnostic accuracy between the same group with/without time constraints, the digital group had a significant performance drop, with the sensitivity dropped by 0.161 and specificity by 0.052 when the time constraint was imposed, whereas the AI-assisted group showed less fluctuation, as shown in Fig. 3c. The detailed experimental results are shown in Supplementary Tables 6 and 7.

**Analysis of false results**. We have performed a thorough analysis of the deep learning model to further improve the pathologists' confidence in the AI system. As shown in Fig. 2c, we listed eight common failure patterns in the daily gastric dataset. The false negative (missed) cases included a well-differentiated adenocarcinoma case (Fig. 2c(i)) and an early atypical signet ring cell carcinoma case involving only the mucosa (Fig. 2c(ii)). Intramucosal well-differentiated adenocarcinoma is morphologically similar to dysplasia and has not yet caused structural disturbances and stromal desmoplasia. For the signet ring cell carcinoma case, the cancer cells were very limited. Apparently, malignancies with minimal structural disturbances in the stroma risk being overlooked. In addition, there were two situations where overdiagnosis might occur (more false positive cases are illustrated in Supplementary Fig. 7). One of them was due to poor image quality, which was related to poor slide preparation, such as section folds (Fig. 2c(viii)), knife marks, and overstaining (Fig. 2c(v)). Poor images also occurred during the digitization stage, for example, poor focus caused by the scanner. These issues may be alleviated with a better data augmentation technique or slide normalization. The second issue was that some lesions were cancer mimickers. For example, mucus extravasation resembled mucinous adenocarcinoma (Fig. 2c(iii)). A correct diagnosis was easier for human pathologists when the slides could be reviewed repeatedly by switching from low to high magnifications. The other case with aggregates of foamy histiocytes in the lamina propria resembled signet ring cell carcinoma (Fig. 2c(iv)), which was again better reviewed by human pathologists under ×40 magnification. Inflammatory necrotic exudates and florid granulation tissue, when there are bizarre endothelial cells and proliferated fibroblasts, could be mistaken as poorly differentiated adenocarcinoma (Fig. 2c(vi) and (vii)). For these cases, human pathologists often needed IHC to help them make a correct diagnosis.

**Multicentre test**. A mature clinically applicable AI assistance system should have robust performance on slides collected from different hospitals. To prove the clinical utility with reproducible sensitivity and specificity of our deep learning model, we tested the performance of our model with slides collected from two other hospitals. We built a multicentre dataset, which included 355 cases (595 slides) from Peking Union Medical College Hospital (PUMCH) and 541 cases (987 slides) from Cancer Hospital, Chinese Academy of Medical Sciences (CHCAMS), to examine whether our model can cope with the variances created by different laboratories, such as different sectioning and staining

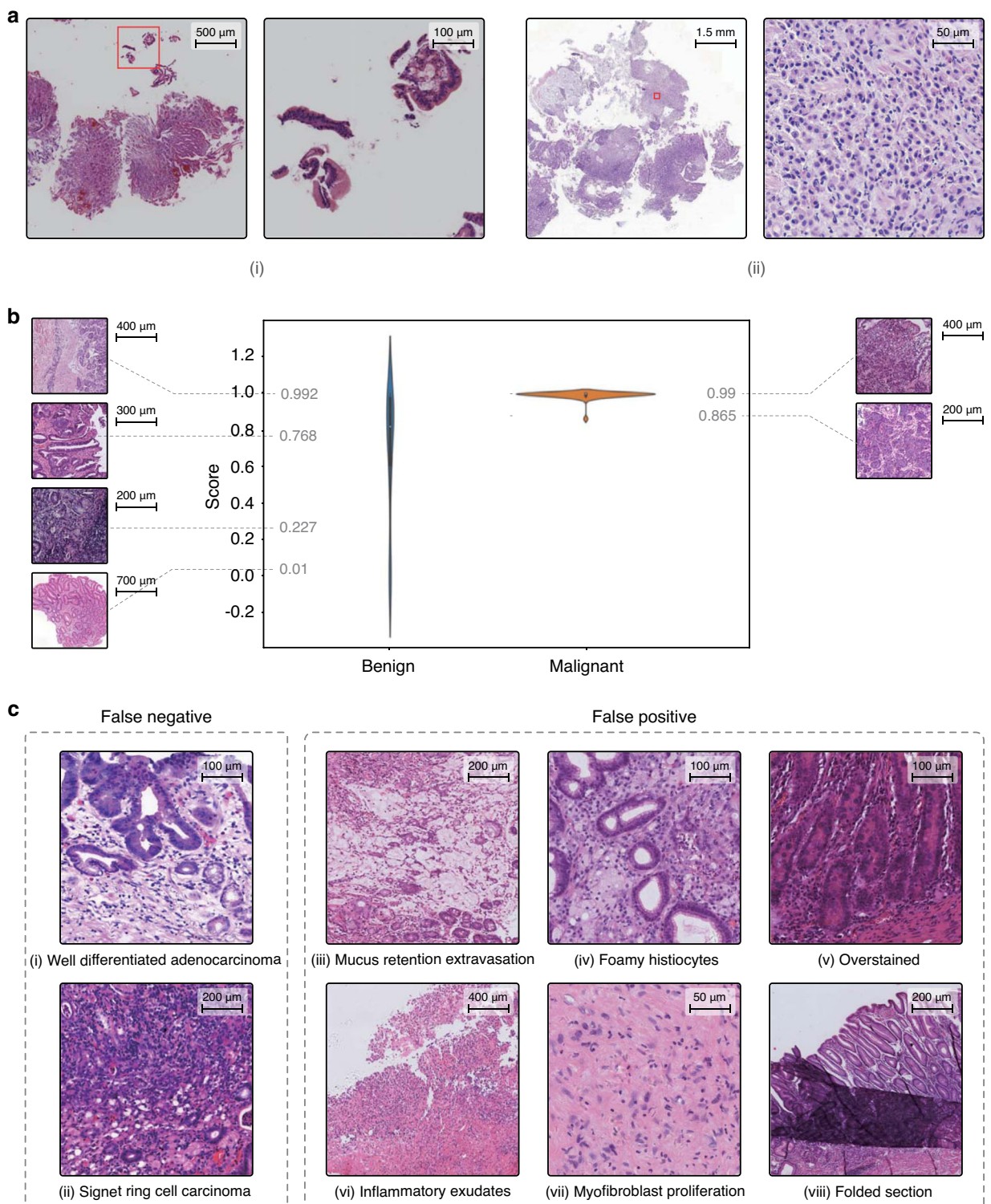

**Fig. 2 Highlights of the deep learning model. a** Two cases detected by the AI assistance system that were initially misdiagnosed by pathologists. **b** Violin plot of the probability distributions for the malignant and benign cases in the IHC dataset. **c** Eight examples of false negative and false positive cases. The experiment was performed five times, and we obtained the same results.

configurations (see Supplementary Fig. 8). The WSIs in the multicentre dataset were produced by the same KF-PRO-005 digital scanner with ×40 magnification. In the comparison of the model performance on the multicentre dataset and the daily gastric dataset, we included 403 WSIs produced by the KF-PRO-005 digital scanner from the daily gastric dataset to

control for the confounding factors. As shown in Fig. 4, the AUC (accuracy, sensitivity, specificity) for the data collected from PUMCH and CHCAMS were 0.990 (0.943, 0.986, 0.937) and 0.996 (0.976, 1.0, 0.968), confirming consistent performance.

In conclusion, we showed that there is a clinical utility for using a deep learning model to improve the diagnostic accuracy

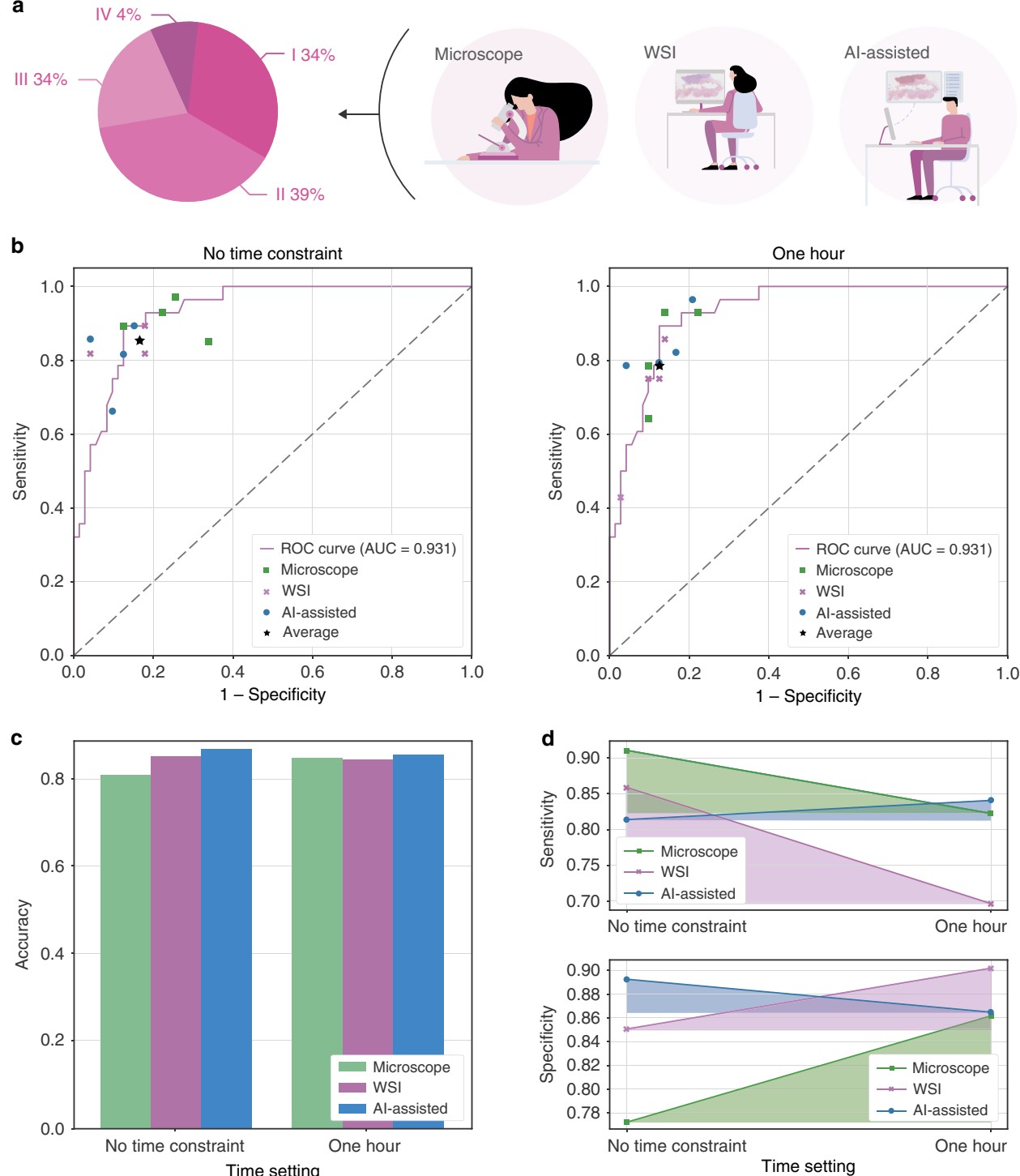

**Fig. 3 Experimental settings and examination results of the performance of the pathology trainees. a** Trainee pathologists were divided into 3 groups to make diagnoses on 100 slides of class I–VI. **b** The model prediction ROC curve and 12 pathologists' performance in the examination. **c** The average diagnostic accuracy of the three groups under different time settings. **d** Diagnostic consistency among different groups.

and consistency of WSIs of gastric cancers. For developing countries with the severe shortage of pathologists, the AI assistance system locates suspicious areas quickly, thus improves diagnostic quality within a limited time frame. On the other hand, for developed countries, the system could help prevent mis-diagnosis. In our practice, to successfully build a clinically applicable histopathological AI assistance system, two factors are essential. The first and foremost goal is to recruit a large number

of WSIs in the training phase covering diverse tumor subtypes with accurate pixel-level annotations under a carefully designed SOP. The annotation process should be monitored constantly by repeated reviews of model predictions to reduce the rates of false negatives and false positives. The second factor is the ability of the AI model to perform pixel-level predictions based on a deep CNN trained with augmented data generated from domain-specific features of histopathology. Our model-building approach can be

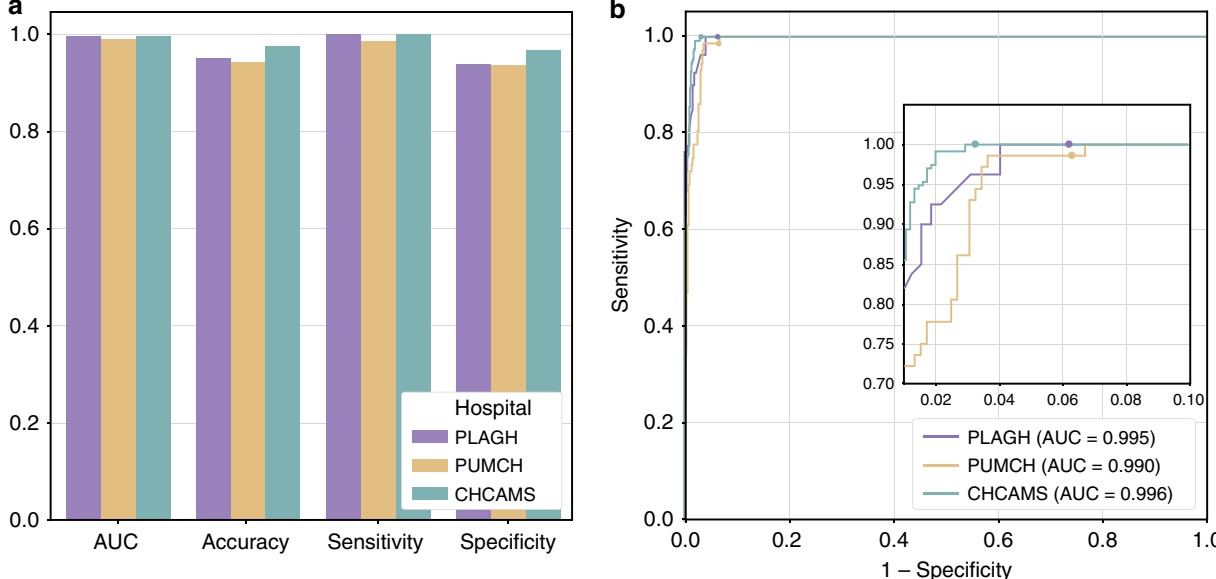

**Fig. 4 Model performance on the multicentre dataset. a** The AUC, accuracy, sensitivity, specificity of the deep learning model on data collected from three hospitals. **b** ROC curves of the model on the multicentre dataset.

applied in the development of histopathological AI assistance systems for a variety of cancers of different organ systems.

## Methods

**Ethical approval**. The study was approved by the institutional review board of each participating hospital (Medical Ethics Committee, Chinese PLAGH; Ethics Committee of PUMCH; Ethics Committee of National Cancer Center/CHCAMS). The informed consents were waived by the institutional review boards since the reports were anonymized. The data used in this research are part of standard-of-care hospital routine.

**Datasets**. The PLAGH dataset was partitioned into six parts: (1) training dataset: contains 2123 WSIs (1391 malignant tumors); (2) training dataset (random forest): contains 737 WSIs (353 malignant tumors); (3) validation dataset: contains 300 WSIs for use for model hyperparameter tuning; (4) internal examination dataset: contains 100 WSIs that were used in the collaboration test; (5) daily gastric dataset: contains 3212 WSIs used in the trial run; (6) IHC dataset: a subset of the daily gastric dataset (99 WSIs) which contains difficult cases that required an immunohistochemistry.

The multicentre dataset contains 595 WSIs from the PUMCH and 987 WSIs from the CHCAMS and Peking Union Medical College.

See Supplementary Tables 8 and 9 for a detailed description and data distribution. We gave an illustration of the test datasets in Supplementary Fig. 9.

**Annotation procedure**. Pixel-level annotations were performed by pathologists from PLAGH on 1391 WSIs. The denotation of malignant tumors for model training was conducted using an iPad-based annotation system. The system interface was shown in Supplementary Fig. 10. We used ThoSlide 2.1.0, a proprietary library, to access the WSIs.

The annotation procedure of a WSI comprised three steps, the initial labeling stage, the verification stage, and the final check stage. A slide was first randomly assigned to a pathologist. Once the labeling was finished, the slide and annotations were then passed on to another pathologist for review. In the final step, a senior pathologist would spot-check 30% of the slides that had passed the first two steps. The algorithm was developed gradually along with the progress of the annotation. To aid the annotation process, we also incorporated a review routine where difficult cases found during the training phase would be sent back for a second-round review.

**Preprocessing**. The annotations we obtained were curves with no specific stroke orders. In the data preprocessing stage, we selected the closed curves and filled in the enclosed areas to obtain pixel-level labels. Outer curves were filled first in the case of nested curves. Otsu's method was applied to the thumbnail of each WSI to obtain the tissue coordinates in the foreground. In practice, the grid search of the thresholding parameter $t$ was performed on the grayscale slide thumbnail to minimize the following function:

$$\sigma_\omega^2(t) = \omega_1(t)\sigma_1^2(t) + \omega_2(t)\sigma_2^2(t), \quad (1)$$

where $\sigma_i^2(t)(i = 1, 2)$ represented intraclass variance, we fixed the weights $\omega_1 = \omega_2 = 0.5$. With the target threshold $t^*$, we could turn the grayscale image into a binary image, marking the tissue area coordinates. The coordinates were then rescaled to the original zoom level to obtain the WSI-level coordinates. We only extracted training patches from coordinates that cover a tissue. During training, the WSIs were split into tiles of $320 \times 320$ pixels in size. We obtained 11,013,286 (malignant: 6,887,275, benign: 4,126,011) training patches with pixel-level annotations.

**Algorithm development**. We built our deep learning model based on DeepLab v3 with the ResNet-50 architecture as its backbone[33]. We also studied the performance of classification (ResNet-50, Inception v3, and DenseNet) and segmentation (U-Net, DeepLab v2, DeepLab v3) models. All models were implemented in TensorFlow[34] using Adam optimizer, the detailed configuration of the training process was listed in Supplementary Table 10.

For the best model (DeepLab v3), the training process took 42.6 h. In the inference stage, we instead used larger tiles of $2000 \times 2000$ pixels and a 10% overlap ratio, by feeding $2200 \times 2200$-pixel tiles into the network while only using the $2000 \times 2000$-pixel central area for the final prediction, to further retain the environment information.

We compared the performance of slide-level prediction approaches including random forest, averaging the top 100, 200, 500, 1000, and 2000 probabilities. To train the random forest, we extracted 30 features (see Supplementary Table 11) from the heatmaps for the training dataset (random forest). The trained classifier was tested on the validation dataset.

The slide-level prediction used in our research was obtained by averaging the top 1000 probabilities.

**AI assistance system design**. The system architecture was illustrated in Supplementary Fig. 11, where we split different system components into microservices. The trained model was served by the containerized TensorFlow Serving[35]. Each worker and TensorFlow Serving pair were bound to a GPU, providing the inference service for the scheduler. Once a client initialized a prediction request, the message was passed to the preprocessing module by the message queue (MQ). Then the effective area of the WSI was cut into tiles and fed into the scheduler. The scheduler managed all the tasks and monitored the workers. When the predictions of all the slide tiles were complete, the postprocessing module merged the tile predictions into one single slide prediction and returned it to the client through the MQ. The client could always send a message to the MQ to query the job progress. Since the communications between the microservices were decoupled by the MQ, and the scheduler manages the tasks independently, our system was designed to be distributable with high scalability. The average inference time of one slide (mean file size of 536.3 MB) was 53.5 and 24.7 s on a server with 4 GPUs and three servers with 12 GPUs. A complete cost analysis for the whole system was given in Supplementary Fig. 12.

**Internal examination**. The settings were there to apply pressure to the trainees to help us understand how one would perform under tremendous pressure. Before the experiment, Z.S. and one trainee (who is not a participant) were asked to perform a

pre-experiment as fast as they could. The slides took Z.S. 40 min to diagnose and took the trainee 52 min. Therefore, the time constraint was set to 1 h. The experiments were carried out in two conditions on the same day. In the morning, each group was asked to finish the 100 test slides within 1 h. After a 3-h break, the pathologists would be reassigned to a different group, and hence, not working under the same setting. In contrast to the morning test, the afternoon test did not have a time constraint. The pathologists were allowed to work at a self-controlled pace. The average years of experience of the attended pathologists were 4.5. For the AI-assisted group, heatmap overlay was displayed over abnormal areas, along with a probability score in the AI assistance system. The heatmap could be turned on and off with a tap on the keyboard space bar. The experiment was performed on MacbookPro 13 with optical mouse. As shown in Supplementary Fig. 13, the trainees gave the diagnosis by clicking the buttons (malignant/benign) on the screen. For the microscope group, the trainees used Olympus BX50.

**Evaluation metrics**. We used slide-level AUC (area under the ROC curve), accuracy, sensitivity, specificity to measure model performance, and accuracy, sensitivity, specificity when comparing with human pathologists. These metrics were defined as follows:

$$\begin{aligned}
\text{Accuracy} &= \frac{N_{TP} + N_{TN}}{N_{TP} + N_{TN} + N_{FP} + N_{FN}}, \\
\text{Sensitivity} &= \frac{N_{TP}}{N_{TP} + N_{FN}}, \\
\text{Specificity} &= \frac{N_{TN}}{N_{TN} + N_{FP}},
\end{aligned} \tag{2}$$

where $N_{TP}$, $N_{TN}$, $N_{FP}$, $N_{FN}$ represented the number of true positive, true negative, false positive, false negative slides, respectively.

**Plots and charts**. All the plots were made using the matplotlib package in Python. The model performance was revealed with both the ROC curve with $1-$specificity as the $x$-axis and sensitivity as the $y$-axis. We adopted bar plots showing the variance of the predictions on time-consecutive data and WSIs from different digital scanners and hospitals. We used line plots to illustrate the internal examination result and to compare performance between different groups. The color fill below the lines serves the purpose of making the visual variation clearer. To study the IHC dataset, we gave a violin plot. The violin plot combined the traditional boxplot with a kernel density estimate (KDE). The KDE gave a rough estimation of the underlying data distribution. The median value was represented by a white dot in the middle. The center thick black bar was the interquartile range, while the thin black line showed the maximum and minimum adjacent values. We used the violin plot to show the prediction distribution from the model, grouped by two classes (malignant and benign).

**Reporting summary**. Further information on research design is available in the Nature Research Reporting Summary linked to this article.

## Data availability

The data that support the findings of this study are available on request from the corresponding authors (H.S. and S.W.). The data are not publicly available due to hospital regulations.

## Code availability

The training code base for the deep learning framework is available at: https://github.com/ThoroughImages/NetFrame. This framework is general and can be applied to other organs. The core components of the inference system are available at: https://github.com/ThoroughImages/PathologyGo.

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

## Acknowledgements

The authors would like to thank Siqi Zheng, Jingsi Chen, Hainan Lu, Xiang Gao, Lang Wang, Lichao Pan, Fangjun Ding, Yao Lu, Li Chen, Daling Liu, and Yuxi Wang at Thorough Images for data processing and helpful discussions. This work is supported by National Natural Science Foundation of China (NSFC) No. 61532001, CAMS Innovation Fund for Medical Sciences (CIFMS) No. 2018-I2M-AI-008, Medical Big Data and Artificial Intelligence Project of the Chinese PLA General Hospital, Tsinghua Initiative Research Program Grant No. 20151080475, and Beijing Hope Run Special Fund of Cancer Foundation of China No. LC2017A07.

## Author contributions

Z.S., S.Z., W.Z., S.W., W.X., and H.S. proposed the research, Y.H., L.S., J.Y., X.G., W.J., Z.W., X.C., X.D., J.L., C.Y., Z.S., and H.S. performed the WSI annotation, S.Z. and W.Z. led the multicentre study, Z.S. and S.W. conducted the experiment, C.K., C.L., Z.S., G.X., and Y.W. wrote the deep learning code and performed the experiment, Z.S., C.K., S.W., X.Z., D.W., and R.C.D. wrote the paper, H.S. and W.X. reviewed the paper.

## Competing interests

X.Z. is the founder of Thorough Images. S.W. is the co-founder and chief technology officer (CTO) of Thorough Images. C.K., C.L., Z.S., G.X., Y.W. are algorithm researchers of Thorough Images. All remaining authors have declared no conflicts of interest.
