## [Peer Review File · Nature Communications]

Reviewers' Comments:

Reviewer #1:

Remarks to the Author:

The manuscript by Song et al aims to present a 'clinically applicable histopathological diagnosis system for gastric cancer detection using deep learning'. The motivation for this work seems to stem from "a critical shortage of anatomical pathologists both nationally and globally, which has created overloaded workforces where diagnostic accuracy being affected".

I have some major concerns regarding the manuscript. First and foremost, the manuscript is extremely difficult to understand due to poor use of the English language. Many sentences, I cannot understand despite reading them several times, as they simply do not make sense. This manuscript needs to be edited for content by a native speaker.

In order to assess the validity of the approach the authors have undertaken, the authors need to provide full details of the 1400 cases they included in the training set. It is insufficient to specify these as 958 surgical specimens (908 malignancies) and 542 biopsies (102 malignancies). A list of diagnoses according to the 5th (!) ed WHO for tumours of the digestive tract with n and % and ICD code/snomed for the benign diagnosis needs to be provided at a minimum, further information on how the slides from resections were selected. The categories of cancer subtypes as provided in Table S2 are neither conform with the current 5th ed WHO nor the 4th ed. WHO classification the authors refer to. Information on tumour stage for the resection specimens is needed as well. The purpose of the 'trial run with the daily gastric dataset' and what authors mean by 'biweek' is not been explained clearly in the manuscript. A flowchart of what happened with what slides in which order could be helpful here. The authors report an area under the curve of 0.986, but fail to provide information about the comparator to assess this.

I am very concerned about the quality of the input into the AI system e.g. the capability of the human observer to classify gastric cancer. I read on page 7 "we found two missed cases that were overlooked in the initial reports and caught by the AI assistance system" and reasons for having been missed by the pathologists are given as "cancer cells were limited in number" or "tumour cells were scattered under normal epithelium and only better visible under high magnification". "These kinds of misdiagnoses were uncommon but possible, especially when a case was read in haste, such as the last case of the day or slides read while multitasking" on page 11 the authors say "apparently, malignancies with minimal structural disturbances in the stroma ran the risk of being overpassed' (I interpret 'overpassed' as missed) – I would consider all of these arguments to justify why a pathologists missed a diagnosis as unacceptable in a routine practice and find it worrying if someone with such relative low quality of pathology diagnosis trains a deep learning algorithm. The quality of the images in figure 2 is not great, but if 'false positive' means that the AI systems classifies them as tumour but the human observer did not, I remain to be convinced that this is true by better HE quality or IHC.

Images shown in Figure S3 labelled as false positive are not always accurately labelled. I assume that the authors mean ectopic or heterotopic pancreas instead of translocated pancreas. Although quality is suboptimal and IHC is not provided, I would think that some of these images are indeed showing cancer cells e.g. are not false positive.

Then, there appears to be several other datasets, one of them called IHC dataset. It is not mentioned what IHC was used for what purpose and how this was integrated into the deep learning process. No details are provided about these slides/datasets, how they were selected, what diagnoses/malignant phenotypes were present in the different sets etc.

What was the purpose of the time constraint experiment and who determined which slides are to be used for this purpose? What workstation did the human observers use as this will have influence on the time it needs to assess a slide?

Have the authors tested model performance comparing intestinal type (tubular adenocarcinoma) versus poorly cohesive carcinoma?

Reviewer #2:

Remarks to the Author:

This paper is about the clinical test-use of AI to assist gastric cancer diagnosis. A large study with over 5000 whole-slide images (WSI), three hospitals and 24 pathologists is reported. The main themes of the paper are:

- data collection and labeling
- AI system description
- Test of AI system

The main findings are that the AI system performs at near 100% sensitivity and 80% specificity in a robust manner (3 different scanners, 3 hospitals). The second finding is that the AI diagnosis system helps pathologist when they can see an overlay of the AI's segmentation.

Overall the paper is well written and the findings are interesting. The large number of slides used for both training and testing make me confident that the reported performance should generalize to other settings. The novelty of the AI system is not high, however, the protocols for testing it are generally well thought out. So I would recommend it for publication.

Some more details. comments and questions:

1: data collection and labeling

In many large scale studies, pixel-level labels (i.e. for malignant cases, the tumor is carefully traced) are lacking and only the diagnosis is available as a label, resulting in the need of multi-instance learning schemes to train a ML model. And conversely, studies with precise labels are usually of a smaller scale. What makes this paper interesting is that the authors labeled 1391 malignant slides at the pixel-level using a iPad app with a stylus and a small army of 12 senior pathologists.

However, it is unclear how the iPad labeling app was developed or whether it is an off-the-shelf product. Also, the labels were double checked and 30% were spot-checked by the most senior pathologist. Usually, slides should be labeled independently by 2 or more pathologists and then the labels should be discussed. The authors should explain why they think their protocol is sufficient.

2: AI system

The system used is not novel and the author do not report on any original tuning method. The model used is Google's Deeplab-v3 atrous CNN which is specialized for segmentation tasks. As far as I can tell, it worked out-of-the-box for them. The authors also report a parallel system architecture with multiple GPUs that seems to be hosted on the Google cloud via TensorFlow-Serving (although this is not made completely clear).

Timing numbers for training and inference are not reported. It would be quite useful to know those to assess the applicability of this approach. In particular, how long does it take to process a slide ?

3: Tests of AI system

During inference, a slide was classified as malignant if the mean over the 1000 largest pixel-level probabilities was over a certain threshold. Since there are myriad ways to evaluate the result of AI inference, this particular choice should be better motivated. The AUC is used as a metric, which is the usual way. The authors, however, also used sensitivity, specificity and accuracy. It is not clear how those are obtained (for example, is the sensitivity set, and the corresponding specificity obtained ?).

It is good to test slides from different institutions to show robustness of the model to variations in tissue preparation. However, it is not very clear whether the variations between the selected hospitals are really significant. No examples are shown to compare staining, etc.

The real-world test on 12 pathologist trainee is novel as far as I know, and shows that the pathologist perform more consistently under time constraints. However, I have to say that the

number of pathologist for such a study is on the small side (12) and the variations among them may make statistics such as 'average increased accuracy using AI' a bit weak.

Reviewer #3:

Remarks to the Author:

This paper describes a system based on convolutional neural networks (DeepLab V3) to segment tumor in gastrointestinal tissue samples.

The system was trained using a fairly large set of slides with manual annotations from a single center.

Validation is done both on an internal dataset with slides scanned with three different scanners, and on an external dataset.

Furthermore, comparison with performance of pathologists on a small subset of the dataset (n=100) is presented.

The authors claim that the presented method is applicable to routine in diagnostics.

Major comments

* Authors claim that "a deep learning model should be able to sustain a thorough test with substantial number of slides". I am not convinced this criterion makes a deep learning system clinically applicable per se, as in principle any system can be run on a large number of slides. Is this a statement about efficiency of computational time? Or do the authors refer to the validation of a deep learning system, rather than its properties? The authors also say that "none met all these criteria", but no comparison is made with the recent paper from Campanella et al., Nature Medicine, which was presented as a clinically applicable system.

* Sensitivity should be near 100%. What is the practical implication of this? Shouldn't sensitivity be 100% to avoid missing any cancers?

* The training dataset is highly unbalanced towards malignant cases (1:9), the authors should explain how they took this into account during model development, and whether this could be a possible cause of suboptimal specificity in some settings.

* Slides difficult to diagnose were annotated as "ignore". I wonder whether this is correct, and how the system would perform on difficult slides, or difficult regions of slides. No visual examples are provided for these cases. A similar comment on associating "poor quality" regions to benign tissue. Were not any poor quality tumor regions in the training set?

* In the reported ROC curves, there is a substantial difference in performance across weeks, which seems to be increasing over time. How can this be explained? This also shows that the best performance was achieved on slides scanned with the same scanner used in the training set, which makes the system not completely robust to scanning differences.

* I find one of the reported cases (page 7) interesting. I understand that cancer cells can be missed, but it's strange to have a patient that undergoes surgery when no cancer was found in the biopsy.

* Color augmentation techniques used in this system might be suboptimal. One indication is in Figure 2(c), where overstained patches are reported as a cause of tumor false positives. Since stain variation is a well-known problem in pathology, a clinically applicable system should be capable of dealing with such a variation in a robust way. However, the authors state in the paper that a better data augmentation or stain normalization might be needed, giving the idea that this system might not be clinically applicable, yet.

* There is a slight improvement of sensitivity using AI from no time constraints to one hour, and a slight drop of specificity, but in general it seems that the analysis under the microscope gives the best performance, looking at the points in the ROC curves. So this does not give a message in favour of using the AI system.

* I see that the time it takes pathologists to do the study, even without time constraints, is often less than 60 minutes, which makes me wonder about the actual difference in the two settings, and what can be the cause of different results obtained.

* The paper contains a several details as well as lack of relevant information. For example, the authors mention that the system was implemented in an iPad and that the Apple Pencil was used, and also explain how the TensorFlow Serving system works, but do not mention basic information like the type of staining used, which I guess being H&E, it can be deduced by the figures, but not explicitly mentioned.

* The authors emphasize the fact that segmentation is used, but they do not show examples of manually annotated samples, or examples of heatmaps / segmentation maps produced by the algorithm. Furthermore, they claim that segmentation gives a more detail-rich prediction at pixel level. Note that this can also be achieved via patch classification and sliding windows, when for example a stride=1 is used. Since a single global score is obtained from all pixel-level predictions, it is not clear what can be gained using a segmentation method. No comparison with other approaches showing the superiority of the chosen approach is reported. Furthermore, no statistical analysis was performed, not even with different settings of the used model architecture.

* The technique to combine pixel-level predictions is just mentioned as the "top 1,000 probabilities", but it is not justified or compared with other techniques. Furthermore, it is not explained why the value 1,000 is used, and whether this is tuned to account for small tumor regions detected by the algorithm.

* Accuracy is often reported as one of the performance metrics. When the validation set is unbalanced, accuracy is not very informative, and does not add much to AUC and Sensitivity/Specificity. Since the authors perform AUC analysis, it is not clear what operating point was picked to report values of sensitivity and specificity.

Responses to Reviewer #1:

Thank you very much for your insightful comments on our manuscript. We have revised our manuscript according to your comments. The responses are listed below.

Point 1: *Many sentences, I cannot understand despite reading them several times, as they simply do not make sense. This manuscript needs to be edited for content by a native speaker.*

Reply: Thank you for your suggestion. The manuscript has been edited by Nature Author Service for proper English language, grammar, punctuation, spelling, and overall style. We believe the use of language is much improved in this revised manuscript.

Point 2: *A list of diagnoses according to the 5th (!) ed WHO for tumours of the digestive tract with n and % and ICD code/snomed for the benign diagnosis needs to be provided at a minimum, further information on how the slides from resections were selected. The categories of cancer subtypes as provided in Table S2 are neither conform with the current 5th ed WHO nor the 4th ed. WHO classification the authors refer to. Information on tumour stage for the resection specimens is needed as well.*

Reply: The clinical utility of our AI model is to assist pathologists in prioritizing their time in providing heatmap cues to pathologists for them to confirm or refute cancer diagnosis. The training dataset is a product of diligent pixel-level annotation by pathologists on a large number of whole slide images (WSIs). We used many resection specimens of gastric cancers, especially in the training dataset. That is because surgical sections cover various cancer subtypes and bring with more annotated training patches. As for the case selection in the testing stage, we included all gastric adenocarcinomas from June 2017 to August 2017. We re-examined all the cases and unified terminology in accordance with the 4th edition of the WHO Classification of Tumours of the Digestive System. Supplementary Tables S8, S9, and the pie chart of data distribution in Figure 1 has been updated accordingly.

Point 3: *The purpose of the 'trial run with the daily gastric dataset' and what authors mean by 'biweek' is not been explained clearly in the manuscript. A flowchart of what happened with what slides in which order could be helpful here.*

Reply: Thank you for your suggestion. We have added a new flowchart in

Supplementary Figure S8 in the revised manuscript to explain the test process in more detail.

Point 4: *The authors report an area under the curve of 0.986, but fail to provide information about the comparator to assess this.*

Reply: In the revised manuscript, we have listed the performance comparison between different deep learning models, including ResNet-50, Inception v3, DenseNet, U-Net, DeepLab v2, and DeepLab v3 (see Supplementary Table S4). We have also included the performance of different slide-level predictors in Supplementary Table S3.

Point 5: *I read on page 7 “we found two missed cases that were overlooked in the initial reports and caught by the AI assistance system” and reasons for having been missed by the pathologists are given as “cancer cells were limited in number” or “tumour cells were scattered under normal epithelium and only better visible under high magnification”. “These kinds of misdiagnoses were uncommon but possible, especially when a case was read in haste, such as the last case of the day or slides read while multitasking” on page 11 the authors say “apparently, malignancies with minimal structural disturbances in the stroma ran the risk of being overpassed’ (I interpret ‘overpassed’ as missed) – I would consider all of these arguments to justify why a pathologists missed a diagnosis as unacceptable in a routine practice and find it worrying if someone with such relative low quality of pathology diagnosis trains a deep learning algorithm.*

Reply: We fully understand your concerns about the quality of our pathology service. In the case illustrated in Figure 2a(i), less than ten atypical cells were present in a tiny break-off tissue fragment. The attending pathologist who made the diagnosis was a GI specialist. She noticed the presence of scanty atypical cells, but due to the scantiness of atypical cells, she could not make a definite diagnosis. She requested a recut, but the lesion area became even smaller on the recut slide. Although a negative diagnosis was made, she made a comment that a small number of atypical cells were present, and malignancy cannot be completely excluded. On OGD, the gastroenterologist saw a malignant-looking ulcer. Even without a definite pathological diagnosis, surgical intervention was still necessary. The examination of the gastrectomy specimen revealed a poorly cohesive cancer. For another case in Figure 2a(ii), the misdiagnosis was made due to the quick diagnosis without higher magnification and IHC confirmation.

The reporting process of PLAGH consists of three steps:

(1) Trainee pathologists read all slides and write primary diagnosis reports.

- (2) Attending pathologists review all the slides and the reports written by trainee pathologists.
- (3) For challenging cases, attending pathologists consult chief pathologists for a final verdict.

Each day, there are 500 new cases on average at the Department of Pathology, PLAGH, corresponding to around 1,500 H&E and 1,500 IHC stained slides. There are 50 pathologists at PLAGH, including 20 trainees and 30 senior pathologists. The average working time for the pathologists is 10 hours a day. Due to the severe shortage of pathologists in China, the current workload has already stretched pathologists to their limits. As a result of the limited diagnostic hours, each slide is diagnosed in about four minutes.

As human beings, we are fallible, including intelligent pathologists. We are subject to failures due to momentarily loss of our focus, especially in confounding situations, i.e., fatigue, emotions, rushing, multitasking, etc. In the practice of medicine, pathological diagnosis by human pathologists has the highest specificity and sensitivity when it comes to distinguishing benign lesions from malignancies. However, the advances of AI systems have clearly demonstrated their superiority in objectivity and consistency.

For comparison, in the labeling process, each slide underwent three rigorous reviewing stages. A slide was first randomly assigned to a pathologist. Once the labeling was finished, the slide and annotations were then passed on to another pathologist for review. In the final step, a senior pathologist would random-check 30% of the slides that had passed the first two steps. The average annotation time for each slide in the first stage was 40 minutes, which is ten times the diagnostic time spent in the daily working environment.

Point 6: *The quality of the images in figure 2 is not great, but if 'false positive' means that the AI systems classifies them as tumour but the human observer did not, I remain to be convinced that this is true by better HE quality or IHC. Images shown in Figure S3 labelled as false positive are not always accurately labelled. I assume that the authors mean ectopic or heterotopic pancreas instead of translocated pancreas. Although quality is suboptimal and IHC is not provided, I would think that some of these images are indeed showing cancer cells e.g. are not false positive.*

Reply: Figure S3 in the old manuscript was re-prepared, and legends were revised now presented in Supplementary Figure S6. The two challenging cases perhaps that we failed to convince you are (c) and (e). For the case - (c), sloughing glandular cells from foveolar epithelium simulates signet ring cells, but if one has a chance to look at the whole slide, it is not difficult to comprehend. Also, the endoscopic finding did at the last patient follow-up in

Mar. 2019 showed no evidence of malignancy. The patient is asymptomatic and in good health condition. For the case (e), it is a gastric ulcer with florid granulation tissue, and we performed IHC with pan CK, CK8&18, and CD34. The following photomicrographs are the IHC stained slides:

Pan CK, granulation tissue area negative.

CK8&18, granulation tissue negative

CD34 highlights vessels and some fibroblastic cells

The judicious use of IHC is an indispensable skill that pathologists are frequently sought for in selected numbers of cases. It is commonplace in our daily practice. Our AI model can only read HE stained slides at this phase. As shown in Figure 2 in the violin plot, the probability distributions for benign lesions are widely spread. In this regard, a small number of benign lesions could be mistaken as malignant based on probability prediction by our AI model.

Point 7: Then, there appears to be several other datasets, one of them called IHC dataset. It is not mentioned what IHC was used for what purpose and how this was integrated into the deep learning process. No details are provided about these slides/datasets, how they were selected, what diagnoses/malignant phenotypes were present in the different sets etc.

Reply: The IHC dataset was generated from the daily gastric dataset. It contains a proportion of cases on which additional IHC tests were performed. We have revised the statements in the main manuscript and provided detailed information of the IHC dataset in Supplementary Tables S8 and S9.

Point 8: What was the purpose of the time constraint experiment and who determined which slides are to be used for this purpose? What workstation did the human observers use as this will have influence on the time it needs to assess a slide?

Reply: The experiment was conducted to simulate the real working condition

of the trainee pathologists. In their daily practice, trainees are required to be able to read 200 slides within 2-3 hours and write down the primary diagnoses. The internal examination dataset comprised 100 slides, which cover a variety of benign to malignant cases with different degrees of diagnostic difficulty. The slides were chosen by Prof. Huaiyin Shi and Prof. Zhigang Song from the daily gastric dataset. Trainees in the WSI and the AI-assisted groups read the WSIs with MacBook Pro 13. As shown in Supplementary Figure S11, the trainees submitted the diagnosis by clicking either benign or malignancy buttons on the screen. For the microscopy group, the trainees used Olympus BX50.

Point 9: Have the authors tested model performance comparing intestinal type (tubular adenocarcinoma) versus poorly cohesive carcinoma?

Reply: The sensitivities for the detection of tubular adenocarcinoma and poorly cohesive carcinoma were 0.998 and 1.0, respectively. Mixed adenocarcinoma was not included. The result is now mentioned in the revised manuscript on page 7.

Responses to Reviewer #2:

Thank you very much for your insightful and helpful comments on our manuscript. We have revised our manuscript according to your comments. The responses are listed below.

Point 1: *However, it is unclear how the iPad labeling app was developed or whether it is an off-the-shelf product.*

Reply: The iPad-based labelling system was developed by computer engineers from Thorough Images with close collaborations with pathologists from PLAGH. The essence behind the development of this system was to make the labelling process simple, intuitive, and allow for fine controlled drawing. Compared with ASAP, which is a widely adopted labelling app, the iPad-based labelling system has three main advantages. Firstly, instead of clicking on the slides with the mouse, pathologists can draw circles with the Apple Pencil, which is faster, more precise, and user friendly. Secondly, the digital slides are stored on a private cloud with encryption. Pathologists can label the slides anywhere whenever the Internet is available, at home or while traveling. Thirdly, each stroke drawn by the pathologists is saved and uploaded synchronously to the cloud, which effectively reduces the risk of data loss. Although the iPad-based labelling system is an in-house software, we will make it an off-the-shelf product soon.

Point 2: *Also, the labels were double checked and 30% were spot-checked by the most senior pathologist. Usually, slides should be labeled independently by 2 or more pathologists and then the labels should be discussed. The authors should explain why they think their protocol is sufficient.*

Reply: Regarding the labelling process, here we would like to compare the reporting process at PLAGH and the labeling process.

The reporting process of PLAGH consists of three steps:

- (1) Trainee pathologists read all slides and write primary diagnosis reports.
- (2) Attending pathologists review all the slides and the reports written by trainee pathologists.
- (3) For challenging cases, attending pathologists consult chief pathologists for a final verdict.

Each day, there are 500 new cases on average at the Department of Pathology, PLAGH, corresponding to around 1,500 H&E and 1,500 IHC stained slides. There are 50 pathologists at PLAGH, including 20 trainees and 30 senior

pathologists. The average working time for the pathologists is 10 hours a day. Due to the severe shortage of pathologists in China, the current workload has already stretched pathologists to their limits. As a result of the limited diagnostic hours, each slide is diagnosed in about four minutes.

For comparison, in the labeling process, each slide underwent three rigorous reviewing stages. A slide was first randomly assigned to a pathologist. Once the labeling was finished, the slide and annotations were then passed on to another pathologist for review. In the final step, a senior pathologist would random-check 30% of the slides that had passed the first two steps. The average annotation time for each slide in the first stage was 40 minutes, which is ten times the diagnostic time spent in the daily working environment.

The effectiveness of the labeling process was reaffirmed by the performance of our model. We provide here a chart showing the total stroke counts of all the annotations, grouped by the stage, including add, modify and delete operations, as below:

The number of strokes gets smaller each stage, which indicates the increase in label quality at each stage.

Point 3: *Timing numbers for training and inference are not reported. It would be quite useful to know those to assess the applicability of this approach. In particular, how long does it take to process a slide?*

Reply: The training process took 42.6 hours. The average inference time for one slide (mean file size 536.3MB) was 53.5 and 24.7 seconds on a server with 4 GPUs and three servers with 12 GPUs, respectively. These statistics are now

in the revised manuscript, page 17.

Point 4: *The authors, however, also used sensitivity, specificity and accuracy. It is not clear how those are obtained (for example, is the sensitivity set, and the corresponding specificity obtained?).*

Reply: The operating point was picked from the ROC curve with sensitivity near 100% and a high specificity on the validation dataset. In the revised manuscript, we have plotted the operating points of the best model in all ROC curves in Figure 4 and Supplementary Figure S4.

Point 5: *However, it is not very clear whether the variations between the selected hospitals are really significant. No examples are shown to compare staining, etc.*

Reply: PLAGH and PUMCH have adopted H&E staining using Leica AutoStainer XL for many years. CHCAMS is using the new Roche Ventana HE 600 automated stainer. A collage with images from the three hospitals is presented in Supplementary Figure S7. We observed a noticeable color variation from different staining methods.

Point 6: *I have to say that the number of pathologists for such a study is on the small side (12) and the variations among them may make statistics such as 'average increased accuracy using AI' a bit weak.*

Reply: There are only 20 trainees at PLAGH. We agree this can be further improved with an experiment with more pathologists. This is where we will be working on in the future.

Responses to Reviewer #3:

Thank you very much for your valuable comments on our manuscript. We have revised our manuscript according to your comments. The responses are listed below.

Point 1: *Authors claim that "a deep learning model should be able to sustain a thorough test with substantial number of slides". I am not convinced this criterion makes a deep learning system clinically applicable per se, as in principle any system can be run on a large number of slides. Is this a statement about efficiency of computational time? Or do the authors refer to the validation of a deep learning system, rather than its properties? The authors also say that "none met all these criteria", but no comparison is made with the recent paper from Campanella et al., Nature Medicine, which was presented as a clinically applicable system.*

Reply: A thorough test with a substantial number of slides is meant to guarantee that the deep learning system is able to capture the real-world data distribution, and can generalize and stand the test when a large amount of out of sample data is presented.

The deep learning system proposed in Campanella et al., Nature Medicine does not meet all the requirements we define in the manuscript:

- (1) They did not perform a study of the model performance on data created with different brands of digital scanners. Over 80% of the WSIs in their test dataset were digitalized by Leica Aperio AT2, only a portion of the prostate WSIs were by Philips IntelliSite Ultra Fast Scanner.
- (2) They did not demonstrate the advantage of using the AI system in assisting pathologists in the diagnostic process.

Point 2: *Sensitivity should be near 100%. What is the practical implication of this? Shouldn't sensitivity be 100% to avoid missing any cancers?*

Reply: We agree in an ideal situation, the sensitivity of cancer detection by an AI system should be 100% while maintaining high specificity. We will keep on working to improve our AI system to meet that goal. In our study, we found that the work of AI systems and pathologists can be complementary. The level of competence of pathologists can vary. Very occasional cases with marked biopsy and slide preparation artifacts may be missed if IHC study or recut are not performed.

Point 3: *The training dataset is highly unbalanced towards malignant cases (1:9), the authors should explain how they took this into account during model development, and whether this could be a possible cause of suboptimal specificity in some settings.*

Reply: On the case level, the training dataset appears to be highly imbalanced. However, if we break it down into the patch level, in each malignant case, there are still large areas within the case that are benign. On the patch level, the ratio of the number of malignant patches to that of benign is 1.5, which is balanced. To further improve model specificity, we will add more benign cases with various subtypes in future work.

Point 4: *Slides difficult to diagnose were annotated as "ignore". I wonder whether this is correct, and how the system would perform on difficult slides, or difficult regions of slides. No visual examples are provided for these cases. A similar comment on associating "poor quality" regions to benign tissue. Were not any poor quality tumor regions in the training set?*

Reply: To help our model learn from difficult cases, we had advised the annotating pathologists not to use the 'ignore' label too often. Moreover, all the slides with ignored regions were reviewed by Prof. Huaiyin Shi and Prof. Zhigang Song, and they allocated proper labels to the ignored regions where applicable. In total, there were 0.28% (30,854/11,013,286) of the patches with ignored pixels. On the other hand, the model performance on the IHC dataset showed that the system is able to diagnose difficult cases. Please also find below a few samples with 'ignore' annotation for reference.

Also, thank you for pointing out the typo. Poor quality regions were actually assigned the label 'ignore' instead of 'benign'. We have corrected the typo in the revised manuscript.

Point 5: *In the reported ROC curves, there is a substantial difference in performance across weeks, which seems to be increasing over time. How can this be explained? This also shows that the best performance was achieved on slides scanned with the same scanner used in the training set, which makes the system not completely robust to scanning differences.*

Reply: It is a coincidence that the AUC increased across weeks. We have shown slides collected from different digital scanners in Supplementary Figure S3. It showed that except for noticeable differences in color, various scanners reveal different imaging qualities. For instance, the blue channel of Hamamatsu NanoZoomer S360 is lighter than two other scanners. The WSIs produced by Ventana DP200 may contain a few blurred spots occasionally.

Point 6: *I find one of the reported cases (page 7) interesting. I understand that cancer cells can be missed, but it's strange to have a patient that undergoes surgery when no cancer was found in the biopsy.*

Reply: The attending pathologist who made the diagnosis was a GI specialist. She noticed the presence of scanty atypical cells, but due to the scantiness of atypical cells, she could not make a definite diagnosis. She requested a recut,

but the lesion area became even smaller on the recut slide. Although a negative diagnosis was made, she made a comment that a small number of atypical cells were present, and malignancy cannot be completely excluded. On OGD, the gastroenterologist saw a malignant-looking ulcer. Even without a definite pathological diagnosis, surgical intervention was still necessary. The examination of gastrectomy specimens revealed a poorly cohesive cancer. At the time of pixel-level labeling, the biopsy slide was reviewed again with the hindsight knowledge of the resection specimen.

Point 7: *Color augmentation techniques used in this system might be suboptimal. One indication is in Figure 2(c), where overstained patches are reported as a cause of tumor false positives. Since stain variation is a well-known problem in pathology, a clinically applicable system should be capable of dealing with such a variation in a robust way. However, the authors state in the paper that a better data augmentation or stain normalization might be needed, giving the idea that this system might not be clinically applicable, yet.*

Reply: Our model could diagnose most histopathological slides correctly, even with various staining methods. Only a few slides with benign lesions were over-diagnosed. The false positives of over-stained cases occurred in the test phase (only 4 cases). Similarly, pathologists can struggle with overstained slides.

We would like to point out that these false positives did not affect the sensitivity of our AI system. The effect of staining variation could be alleviated by adequate training data and also by data augmentation techniques such as color jittering. The limited over-diagnosed cases are related to the fact that the current color jittering method was done within a conservative range. We are working on better data augmentation or stain normalization techniques to further improve the specificity of our model. In addition, to improve the system's reliability, we are working on the quality assurance module to block poor quality slides from going into the AI model. For those slides with suboptimal quality, although they are still allowed to be fed into the AI model, a pop-up message will be displayed to alert pathologists.

Point 8: *There is a slight improvement of sensitivity using AI from no time constraints to one hour, and a slight drop of specificity, but in general it seems that the analysis under the microscope gives the best performance, looking at the points in the ROC curves. So this does not give a message in favour of using the AI system.*

Reply: The trainee pathologists who participated in the examinations were all new to digital slides and had not practiced with WSIs before. Recent

research* has shown that digital pathology can help pathologists perform equally well, if not better than, with a microscope. However, depending on the individual, it usually takes two weeks to get familiar with viewing WSI on the screen. Once getting used to WSIs, many pathologists preferred digitalized slides to the conventional microscope. Therefore, we attributed the decline in performance with WSIs as a result of unfamiliarity with the new diagnostic mechanism.

When comparing the performance between the WSI and the AI-assisted group where the same digital slides were used with no visual difference, the AI-assisted group outperformed the WSI group in both experiments. The performance gap between the WSI group and the microscopy group, probably due to the unfamiliarity, indicated that the AI assistance system would help the pathologists further as they get more used to working with the system.

* Retamero J A, Aneiros-Fernandez J, del Moral R G. Complete digital pathology for routine histopathology diagnosis in a multicenter hospital network. Archives of Pathology & Laboratory Medicine, 2020, 144(2): 221-228.

Point 9: *I see that the time it takes pathologists to do the study, even without time constraints, is often less than 60 minutes, which makes me wonder about the actual difference in the two settings, and what can be the cause of different results obtained.*

Reply: The report process of PLAGH consists of three steps:

- (1) The trainees read all the slides and write the primary diagnosis reports.
- (2) The attending pathologists review all the slides diagnosed by the trainees.
- (3) For hard cases, the attending pathologists pass the slides to the (associated) chief pathologists for a final decision.

On average, there are 500 new cases at the Department of Pathology, PLAGH, corresponding to around 1,500 H&E stained and 1,500 IHC slides. There are about 50 pathologists at PLAGH, including 20 trainees and 30 senior pathologists. The average working time for the pathologists is 10 hours a day. As a result of the limited diagnostic time, on average, each slide is diagnosed in four minutes.

The experiment was conducted to simulate the working state of the trainees. In the real scenario, trainees need to complete the primary diagnosis of over 200 slides within 2-3 hours. The experiment with time constraints was designed to impose pressure on the trainees. From the results, we can see that the average sensitivity of the trainees was lower under time constraints

without the help of the AI assistance system.

Point 10: *The paper contains a several details as well as lack of relevant information. For example, the authors mention that the system was implemented in an iPad and that the Apple Pencil was used, and also explain how the TensorFlow Serving system works, but do not mention basic information like the type of staining used, which I guess being H&E, it can be deduced by the figures, but not explicitly mentioned.*

Reply: Thank you for pointing this out. We have now included the staining type in the revised manuscript.

Point 11: *The authors emphasize the fact that segmentation is used, but they do not show examples of manually annotated samples, or examples of heatmaps / segmentation maps produced by the algorithm. Furthermore, they claim that segmentation gives a more detail-rich prediction at pixel level. Note that this can also be achieved via patch classification and sliding windows, when for example a stride=1 is used. Since a single global score is obtained from all pixel-level predictions, it is not clear what can be gained using a segmentation method. No comparison with other approaches showing the superiority of the chosen approach is reported. Furthermore, no statistical analysis was performed, not even with different settings of the used model architecture.*

Reply: Thank you for your suggestion. A selection of annotated slides and prediction heatmaps is now included and shown in Supplementary Figures S1 and S5.

Point 12: *The technique to combine pixel-level predictions is just mentioned as the "top 1,000 probabilities", but it is not justified or compared with other techniques. Furthermore, it is not explained why the value 1,000 is used, and whether this is tuned to account for small tumor regions detected by the algorithm.*

Reply: We compared the performance of slide-level postprocessing approaches, including random forest, averaging the top 100, 200, 500, 1000, and 2,000 probabilities. To train the random forest, we extracted 30 features from the heatmaps for the training dataset. The trained classifiers were tested on the validation dataset. The detailed results are shown in Supplementary Table S3.

Point 13: *Accuracy is often reported as one of the performance metrics. When the validation set is unbalanced, accuracy is not very informative, and does not add much to AUC and Sensitivity/Specificity. Since the authors*

perform AUC analysis, it is not clear what operating point was picked to report values of sensitivity and specificity.

Reply: The operating point was picked from the ROC curve with sensitivity near 100% and a high specificity on the validation dataset. In the revised manuscript, we plotted the operating points of the best models in all ROC curves in Figure 4 and Supplementary Figure S4.

Summary of Changes:

Major revision.

1. The manuscript has been edited by Nature Author Service for proper English language, grammar, punctuation, spelling, and overall style.
2. We re-examined all the data in Supplementary Tables S8 and S9 in accordance with the 4th edition of the WHO Classification of Tumours of the Digestive System. The pie chart of data distribution in Figure 1 has been updated accordingly.
3. We have included the performance of different slide-level predictors in Supplementary Table S3. We have also listed the performance comparison between different deep learning models in Supplementary Table S4.
4. We discovered that we made a mistake in calculating the slide number in the IHC dataset. We have corrected all the statistics on page 9, i.e., ... IHC dataset contained 36 surgical specimens (28 malignancy) and 63 biopsies (14 malignancies). Our model achieved an AUC of 0.923 (accuracy: 0.808, sensitivity: 0.976, specificity: 0.684).
5. We explained the test process in more detail by providing a new flow chart in Supplementary Figure S8.
6. The interface of the internal examination system was given in Supplementary Figure S11.
7. To reveal the visual difference of various staining methods, we presented a collage with images from the three hospitals in Supplementary Figure S7.
8. We have shown slides collected from different digital scanners in Supplementary Figure S3.
9. A selection of annotated slides and prediction heatmaps is now included and shown in Supplementary Figures S1 and S5.
10. We have plotted the operating points of the best model in all ROC curves in Figure 4 and Supplementary Figure S4.
11. The source code for the training framework was open sourced at: <https://github.com/ThoroughImages/NetFrame>.

Minor revision.

1. The legends in Supplementary Figure S6 were revised.
2. We fixed some typos in the revised manuscript:
 - (1) Poor quality regions were assigned the label 'ignore' (old version: 'benign') on page 4.
 - (2) ResNet-50 (old version: ResNet-34) on page 17.
3. The apparatus used during the internal examination was mentioned on page 19.

4. The sensitivities for the detection of tubular adenocarcinoma and poorly cohesive carcinoma were listed on page 7.
5. The timing number of the training and inference processes were introduced on page 17.

Reviewers' Comments:

Reviewer #1:

Remarks to the Author:

This reviewer appreciates the extensive work that has gone into preparing a revised version of the manuscript.

From the comments to the reviewer's questions it becomes apparent that a number of procedures and underlying reasoning are specific to the settings in the local Chinese pathology department (staffing, workload etc). The situation in pathology departments in other parts of the world may be different, and other departments may therefore have different needs for an AI system to help with daily routine. I would therefore recommend that the authors include a statement to this effect e.g. how useful they think their developed AI system might be in a different pathology department setting in different parts of this world and in that context, a statement on costs to implement their system into the routine workflow versus assumed savings (costs, manpower, lower number of inaccurate diagnosis) would be valuable to the reader.

Reviewer #2:

Remarks to the Author:

I appreciate the thorough rebuttal. My questions were adequately answered and the manuscript correspondingly revised. Overall I can recommend the manuscript for publication.

Reviewer #3:

Remarks to the Author:

I would like to thank the authors for addressing some of my previous comments.

In general, I find that the method section is still a bit too schematic and lacking details that would be needed to reimplement the proposed method. I do acknowledge that the authors have made their code publicly available. I also find that some important points were not properly addressed. For example, I would have expected an analysis on the implication of a sensitivity < 100% in the clinical settings (question #2), for example in terms of extra work for pathologists to look for true positives rather than only discarding false positives, whereas the authors replied that they are working on improving their system to get 100% sensitivity.

Given the explanation provided by the authors, I am also wondering what is the validity of an experiment with a time constraint of 1 hour, when in practice pathologists have to diagnose this type of case in 4 minutes. Shouldn't this experiment be more in line with reality?

My question #11 touched upon several points, starting from lack of figures on manual annotations and heatmaps, on motivation for using a segmentation model, on the lack of motivation for choosing the methods used in the comparison, and a lack of an actual comparison from the statistical point of view, as well as technical details how each model was trained, applied, etc. The authors only addressed the very first point of this question, meaning adding figures with manual annotations and heatmaps, but ignored the rest of the question.

From the paper and from the answer to question #12, I understand that the random forest classifier was trained using features extracted from the training set after having classified the training set with the trained DeepLabV3 model. I think this approach is not correct, because the CNN can overfit on the training set and predict heatmaps (then used to extract features) that do not represent the ones obtained at test set. The validation set should have been used to train random forest, and an additional subset of the validation set, not used for training the RF, should

have been used to assess the performance.

Responses to Reviewer #1:

Thank you very much for your helpful comments on our manuscript. We have revised our manuscript according to your comments. The responses are listed below.

From the comments to the reviewer's questions it becomes apparent that a number of procedures and underlying reasoning are specific to the settings in the local Chinese pathology department (staffing, workload etc). The situation in pathology departments in other parts of the world may be different, and other departments may therefore have different needs for an AI system to help with daily routine. I would therefore recommend that the authors include a statement to this effect e.g. how useful they think their developed AI system might be in a different pathology department setting in different parts of this world and in that context, a statement on costs to implement their system into the routine workflow versus assumed savings (costs, manpower, lower number of inaccurate diagnosis) would be valuable to the reader.

Reply: Thank you for your suggestion. In the revised manuscript, we added statements on how the system would help pathologists in developing countries like China and developed counties with their diagnostic work in the conclusion.

“For developing countries with the severe shortage of pathologists, the AI assistance system locates suspicious areas quickly, thus improves diagnostic quality within a limited time frame. On the other hand, for developed countries, the system could help prevent misdiagnosis.” (Page 14; Line 5)

Meanwhile, we also gave the cost analysis of the AI assistance system in Supplementary Figure S12.

GPU Number	Server Cost	Processing Capacity / 12 Hours
1	\$4,000	200 Slides
2	\$7,000	400 Slides
3	\$11,000	600 Slides
4	\$14,000	800 Slides

Digital Scanner	
Roche Ventana DP200 (6 slides):	\$80,000
Hamamatsu NanoZoomer S360 (360 slides):	\$150,000

Total Cost	
Small hospital:	\$84,000-\$87,000
Large hospital:	\$161,000-\$164,000

The cost for the digital scanners were estimations of the current market price. The server hardware configuration was: [CPU] Intel Core i7, [Memory] 32GB, [Solid State Disk] 1TB, [Hard Disk Drive] 10TB, [GPU] NVIDIA Tesla P100. The cost of the diagnostic system was not included.

For developing countries, the current workload has already stretched pathologists to their limits. The AI assistance system with sensitivity near 100% would help the pathologists locate the suspicious areas and reduce the workload. For developed countries, the system could act as an analogue to a second opinion from fellow pathologists, thus prevent misdiagnosis.

Based on the test result (~2 / 3000 WSIs) on the daily gastric dataset at PLAGH, the number of inaccurate WSI-level diagnoses prevented by the system was estimated to be 25 to 100 in a year.

Responses to Reviewer #2:

Thank you very much for your positive comments on our manuscript.

Responses to Reviewer #3:

Thank you very much for your insightful comments on our manuscript. We apologize for not adequately answering all the comments in the last revised version of the manuscript. We have further revised our manuscript according to your comments. The responses are listed below.

Point 1: *In general, I find that the method section is still a bit too schematic and lacking details that would be needed to reimplement the proposed method. I do acknowledge that the authors have made their code publicly available.*

Reply: In the revised manuscript, we have listed all the details implementing the proposed deep learning model and its comparators in Supplementary Table S10. In order to promote the future development of pathological AI, we further open-sourced the core components of PathologyGo, the AI assistance system designed for histopathological inference, at <http://github.com/ThoroughImages/PathologyGo>.

Point 2: *I also find that some important points were not properly addressed. For example, I would have expected an analysis on the implication of a sensitivity < 100% in the clinical settings (question #2), for example in terms of extra work for pathologists to look for true positives rather than only discarding false positives, whereas the authors replied that they are working on improving their system to get 100% sensitivity.*

Reply: We fully understand your concern. The sensitivity and specificity were derived under a certain threshold. These statistics reported in the manuscript were calculated under the threshold of 0.92, corresponding to 1, 1, 0 false negative slides for test sets collected from PLAGH, PUMCH, CHCAMS, respectively. The predicted heatmaps for the missed cases were shown here.

PLAGH

PUMCH

Although the probabilities were relatively low, you may still find that the deep learning model correctly located some of the suspicious areas. In real-world applications, cases with probabilities between 0.86 and 0.92 are put into the “suspicious bin” along with the heatmap for further review. Once the pathologist opened the case, the system would pop up a warning of suspicion so that it doesn’t get missed out.

We also performed a thorough analysis on the daily gastric dataset. They were 217 slides (6.8%) in the suspicious bin, among which 1 was malignant and 216 were benign. For the multicenter datasets, there were only 25 (1 malignant; 24 benign; 4.2%) and 23 slides (0 malignant; 23 benign; 2.3%) in the suspicious bin for PUMCH and CHCAMS, respectively.

Point 3: *Given the explanation provided by the authors, I am also wondering what is the validity of an experiment with a time constraint of 1 hour, when in practice pathologists have to diagnose this type of case in 4 minutes. Shouldn't this experiment be more in line with reality?*

Reply: The settings were there to apply pressure to the trainees to help us understand how one would perform under tremendous pressure. Different from the routine diagnosis process with report writing, the pathologists were only asked to determine whether the slides were malignant or not.

Before the experiment, Prof. Zhigang Song and one trainee (who is not a participant) performed a pre-experiment as fast as they could. The slides took Prof. Zhigang Song 40 minutes to diagnose and took the trainee 52 minutes. Therefore, the time constraint was set to one hour. We added this important information in the revised manuscript. (Page 18; Line 17)

Point 4: *My question #11 touched upon several points, starting from lack of figures on manual annotations and heatmaps, on motivation for using a segmentation model, on the lack of motivation for choosing the methods used in the comparison, and a lack of an actual comparison from the statistical point of view, as well as technical details how each model was trained, applied, etc. The authors only addressed the very first point of this question, meaning adding figures with manual annotations and heatmaps, but ignored the rest of the question.*

Reply: Thank you for giving us the chance to improve our manuscript. In the revised manuscript, we make our statement clearer by revealing the advantage of the proposed method over existing ones on three aspects:

1. The proposed method achieved better performance.

As shown in Supplementary Table S4 in the revised manuscript, the DeepLab v3 model outperforms its comparators.

2. The inference time of the segmentation model was significantly shorter than classification models.

In the revised manuscript, we gave the inference time of different deep learning models in Supplementary Figure S2(a). For the classification model, in order to make the result more interpretable, the stride should be small, thus the computing time becomes significantly longer.

3. The predictions of the segmentation model were more interpretable than the classification model.

We showed several heatmaps for both segmentation (DeepLab v3) and classification (Inception v3) models in Supplementary Figure S2(b). Within realistic inference time, we could see that the segmentation model reveals more interpretable predictions.

Point 5: *From the paper and from the answer to question #12, I understand that the random forest classifier was trained using features extracted from the training set after having classified the training set with the trained DeepLabV3 model. I think this approach is not correct, because the CNN can overfit on the training set and predict heatmaps (then used to extract features) that do not represent the ones obtained at test set. The validation set should have been used to train random forest, and an additional subset of the validation set, not used for training the RF, should have been used to assess the performance.*

Reply: Thank you for pointing this out. In the revised manuscript, we constructed another dataset to train the random forest. The detailed data distribution was listed in Supplementary Tables S8 and S9. The model performance was tested on the validation set. We have revised Supplementary Table S3 accordingly.

Summary of Changes:

Major revision.

1. We added statements on how the system would help pathologists in developing countries like China and developed countries with their diagnostic work in the conclusion. Meanwhile, we have also gave the cost analysis in Supplementary Figure S12.
2. We gave the inference time of different deep learning models in Supplementary Figure S2(a). In addition, we showed several heatmaps for both segmentation (DeepLab v3) and classification (Inception v3) models in Supplementary Figure S2(b).
3. In order to promote the future development of pathological AI, we further open-sourced the core components of PathologyGo, the AI assistance system designed for histopathological inference, at <http://github.com/ThoroughImages/PathologyGo>.

Minor revision.

1. We added more information about the internal examination.
2. We constructed another dataset to train the random forest. The detailed data distribution was listed in Supplementary Tables S8 and S9. The model performance was tested on the validation set. We have revised Supplementary Table S3 accordingly.

Reviewers' Comments:

Reviewer #3:

Remarks to the Author:

The authors have positively addressed my comments. Therefore, I think the manuscript is ready for publication in its current form.

Responses to Reviewer #3:

Thank you very much for your positive comments on our manuscript.